# Ginsenoside Re Mitigates 6-Hydroxydopamine-Induced Oxidative Stress through Upregulation of GPX4

**DOI:** 10.3390/molecules25010188

**Published:** 2020-01-02

**Authors:** Gyeong Hee Lee, Won Jin Lee, Jinwoo Hur, Eunsu Kim, Hyuk Gyoon Lee, Han Geuk Seo

**Affiliations:** Department of Food Science and Biotechnology of Animal Resources, College of Sang-Huh Life Science, Konkuk University, 120 Neungdong-ro, Gwangjin-Gu, Seoul 05029, Korea; kyung9642@naver.com (G.H.L.); 486251793@naver.com (W.J.L.); jinwoo910218@naver.com (J.H.); np-gennao@hanmail.net (E.K.); krci-12@daum.net (H.G.L.)

**Keywords:** ginsenoside Re, glutathione peroxidase 4, oxidative stress, neuronal damage, SH-SY5Y cells

## Abstract

Ginsenosides are active components found abundantly in ginseng which has been used as a medicinal herb to modify disease status for thousands of years. However, the pharmacological activity of ginsenoside Re in the neuronal system remains to be elucidated. Neuroprotective activity of ginsenoside Re was investigated in SH-SY5Y cells exposed to 6-hydroxydopamine (6-OHDA) to induce cellular injury. Ginsenoside Re significantly inhibited 6-OHDA-triggered cellular damage as judged by analysis of tetrazolium dye reduction and lactose dehydrogenase release. In addition, ginsenoside Re induced the expression of the antioxidant protein glutathione peroxidase 4 (GPX4) but not catalase, glutathione peroxidase 1, glutathione reductase, or superoxide dismutase-1. Furthermore, upregulation of GPX4 by ginsenoside Re was mediated by phosphoinositide 3-kinase and extracellular signal-regulated kinase but not by p38 mitogen-activated protein kinase or c-Jun N-terminal kinase. Ginsenoside Re also suppressed 6-OHDA-triggered cellular accumulation of reactive oxygen species and peroxidation of membrane lipids. The GPX4 inhibitor (1S,3R)-RSL3 reversed ginsenoside Re-mediated inhibition of cellular damage in SH-SY5Y cells exposed to 6-OHDA, indicating that the neuronal activity of ginsenoside Re is due to upregulation of GPX4. These findings suggest that ginsenoside Re-dependent upregulation of GPX4 reduces oxidative stress and thereby alleviates 6-OHDA-induced neuronal damage.

## 1. Introduction

A hydroxylated analog of dopamine, 6-hydroxydopamine (6-OHDA), has been noted to cause degeneration of both noradrenergic and dopaminergic neurons [1]. 6-OHDA easily enters these types of neurons due to its high affinity for both dopaminergic and noradrenergic transporters [2]. Once transported into neurons, 6-OHDA is readily oxidized and this leads to the formation of reactive oxygen species (ROS), which trigger oxidative stress in dopaminergic neurons and ultimately induce cytotoxicity [3]. Thus, 6-OHDA is popularly adopted to generate an animal model of Parkinson’s disease (PD)—the progressive neurodegeneration of dopaminergic neurons in the substantia nigra. Moreover, the use of 6-OHDA to generate such a cellular PD model is validated by reports that this compound is detected in the human urine and brain of PD patients [4,5,6,7].

Oxidative stress, which results from an imbalance between cellular ROS and the antioxidant defense system, not only triggers severe neurodegenerative diseases including PD but also facilitates disease progression by inducing neuronal cell death [8]. Moreover, the high level of glutathione is not sufficient to protect the cells treated with 6-OHDA, indicating the importance of antioxidant enzymes in neuronal oxidative injury [9]. ROS are generated by normal metabolic processes and removed by many antioxidant enzymes including catalase (CAT), glutathione peroxidase (GPX), glutathione reductase (GR), and superoxide dismutase (SOD). Among these enzymes, GPX4 is emerging as a principal antioxidant enzyme because of its unique ability to reduce phospholipid peroxidation [10]. A recent study reported that GPX4 deficiency causes embryonic death and that mice expressing mutated GPX4 with low enzymatic activity die at postnatal day 18 due to excessive oxidative stress in the brain [11], implying that GPX4 is important to prevent oxidative stress-mediated neuronal damage.

Ginseng has been used as a medicinal herb to modify disease status for thousands of years [12]. Ginsenosides are active components found abundantly in ginseng [12,13]. Among their multiple biological activities, many ginsenosides (Figure 1), including Rg1, Rb1, Rb2, and Rd, have antioxidant properties [14,15,16,17,18,19]. In particular, ginsenoside Rb1 reduces cerebral ischemia-induced injuries and oxidative stress in mouse spinal cords and aged mice, respectively, by regulating antioxidant signaling pathways [16,17]. Ginsenosides Rb2 and Rd attenuate oxidative stress and cell death triggered by glutamate and amyloid-β (Aβ) in hippocampal neuronal cells, respectively [18,19]. Similarly, ginsenosides Rg1 and Rg3 protect neuronal cells from oxidative stress-triggered apoptosis and transient focal cerebral ischemia, respectively [15,20]. Based on the antioxidant effects of multiple ginsenosides, we hypothesized that ginsenoside Re has a similar activity. However, the antioxidant mechanism of ginsenoside Re in the neuronal system has not been fully explored, although it was recently reported to have such an effect in Aβ-challenged SH-SY5Y through the Nrf2 signaling pathway [21]. Thus, we tested the effects of ginsenoside Re in an in vitro model of PD induced by 6-OHDA and explored the underlying mechanisms.

## 2. Results

### 2.1. Ginsenoside Re Attenuates 6-OHDA-Triggered Cellular Damage

The SH-SY5Y cell line is widely used as a cellular model for the degeneration of dopaminergic neurons induced by neurotoxin [22,23]. In previous experiments, we measured the effect of 6-OHDA on SH-SY5Y cell viability (Figure 2A). Cell viability decreased when the concentration of 6-OHDA was increased. Since our purpose was to demonstrate the protective effect of ginsenoside Re on 6-OHDA cell survival, SH-SY5Y cells were exposed thereafter to 20 µM of 6-OHDA, a treatment that decreases cell viability up to 30%. Next, we determined the cytotoxic effect of ginsenoside Re. When SH-SY5Y cells were exposed to various concentrations of this compound for 24 h, cell viability was not affected by ginsenoside Re even at the highest dose (200 μM) (Figure 2B). Thus, we assessed the effects of ginsenoside Re on 6-OHDA-triggered cellular damage by determining the level of lactate dehydrogenase (LDH) released into the medium and 3-(4,5-dimethylthiazol-2-yl)-2,5-diphenyltetrazolium bromide (MTT)-based cell viability in SH-SY5Y cells. 6-OHDA significantly increased LDH release into the medium as reported previously [3]. However, this effect was attenuated in the presence of ginsenoside Re (Figure 3A). Treatment with 10–100 μM of ginsenoside Re significantly reduced 6-OHDA-triggered LDH release. MTT-based assessment of cell viability confirmed that ginsenoside Re elicited a neuroprotective effect in 6-OHDA-treated SH-SY5Y cells (Figure 3B).

### 2.2. Ginsenoside Re Upregulates the Expression of GPX4

Oxidative stress is implicated in 6-OHDA-triggered cell damage [3]; thus, we examined the effects of ginsenoside Re on the expression of the antioxidant proteins SOD1, GR, CAT, GPX1, and GPX4 in SH-SY5Y cells. Treatment applied induced only changes at the mRNA level of GPX4 (Figure 4). The level of *GPX4* mRNA was significantly increased by treatment with 25 μM ginsenoside Re, and this upregulation reached a maximum of nearly three-fold after 9 h (Figure 5A). Similarly, treatment with ginsenoside Re for 9 h dose-dependently enhanced the expression of *GPX4* mRNA in SH-SY5Y cells (Figure 5B). To determine whether enhanced *GPX4* mRNA expression is followed by the expression of GPX4 protein, the protein level of GPX4 was determined in ginsenoside Re-treated SH-SY5Y cells. Ginsenoside Re enhanced the protein level of GPX4 in a time- and dose-dependent manner (Figure 5C,D).

### 2.3. Ginsenoside Re Reduces 6-OHDA-Induced Oxidative Stress

6-OHDA causes neurotoxicity by inducing oxidative stress [24]; thus, the effect of ginsenoside Re on 6-OHDA-triggered oxidative stress was evaluated. A significant increase in DCF fluorescence, an indicator of ROS, was observed in SH-SY5Y cells treated with 6-OHDA. However, this increase in ROS production was almost completely abolished in cells pretreated with ginsenoside Re for 9 h, indicating that this compound has antioxidant activity (Figure 6A,B).

Ginsenoside Re upregulates the expression of GPX4 in SH-SY5Y cells, and GPX4 suppresses phospholipid peroxidation, which causes cellular damage [9]. Thus, we evaluated the effect of ginsenoside Re on 6-OHDA-triggered lipid peroxidation. 6-OHDA significantly increased lipid peroxidation as assessed by a fluorescent probe C11-BODIPY (581/591) [25]. Preincubation with 25 μM ginsenoside Re for 9 h significantly attenuated the 6-OHDA-mediated increase in the oxidized BODIPY level (Figure 6C,D). These results suggest that ginsenoside Re-mediated upregulation of GPX4 is associated with the neuroprotective effects of this compound against oxidative damage triggered by 6-OHDA.

### 2.4. Ginsenoside Re Regulates the Expression of GPX4 via the Phosphoinositide 3-Kinase/Akt and Extracellular Signal-Regulated Kinase Cascades

Because diverse ginsenoside derivatives treatments [26,27,28,29] activate the phosphoinositide 3-kinase (PI3K)/Akt signaling pathway as a cell survival signaling pathway [30], we assessed whether it is involved in cell survival of 6-OHDA treated cells. To examine the signaling pathways activated by ginsenoside Re the effects of this compound on the phosphorylation of PI3K/Akt, p38, extracellular signal-regulated kinase (ERK), and c-Jun N-terminal kinase (JNK) were analyzed. SH-SY5Y cells were treated with 25 μM ginsenoside Re for the indicated duration then Western blots were performed with phospho-specific antibodies. The total level of each kinase was determined in parallel immunoblots using the respective kinase-specific antibodies. Ginsenoside Re activated the PI3K/Akt and ERK cascades but not the JNK and p38 signaling pathways (Figure 7A,C). Activation of PI3K/Akt and ERK peaked after ginsenoside Re treatment for 6 h and 9 h, respectively.

To further confirm the signaling cascades involved in the upregulation of GPX4 by ginsenoside Re, we inhibited the signal pathways of PI3K/Akt, ERK, p38, and JNK using specific inhibitors. Treatment with PI3K inhibitor LY294002 and ERK inhibitor PD98059 significantly suppressed the ginsenoside Re-stimulated increase in the GPX4 protein level, suggesting that the PI3K/Akt and ERK cascades are linked with upregulation of GPX4 by ginsenoside Re (Figure 7B,D). By contrast, inhibition of p38 and JNK using SB203580 and SP600125, respectively, did not affect the ginsenoside Re-triggered upregulation of GPX4 protein, suggesting that the p38- and JNK-mediated cascades are not associated with upregulation of GPX4 by ginsenoside Re.

### 2.5. The GPX4 Inhibitor (1S,3R)-RSL3 Reverses the Effect of Ginsenoside Re against Oxidative Stress-Caused Neuronal Injury

To assess the consequence of GPX4 upregulation in ginsenoside Re-treated SH-SY5Y cells, LDH release was assayed. Treatment of SH-SY5Y cells with 6-OHDA for 24 h dramatically increased LDH release into the culture medium (Figure 8A). Ginsenoside Re significantly attenuated 6-OHDA-triggered LDH release; however, a GPX4 inhibitor (1*S*,3*R*)-RSL3 almost completely abolished this effect. This finding was further verified by assessment of cell viability (Figure 8B). These results indicate that GPX4 is a primary factor in the neuroprotective action of ginsenoside Re against cellular damage triggered by 6-OHDA.

## 3. Discussion

In the present report, we demonstrated that the ginseng-derived medicinal compound ginsenoside Re upregulated GPX4 expression in a time- and dose-dependent manner. This upregulation involved the PI3K/Akt and ERK signaling cascades. Furthermore, ginsenoside Re rendered SH-SY5Y cells resistant to 6-OHDA-triggered cellular damage, while inhibition of GPX4 abolished the neuroprotective effects of ginsenoside Re.

Upregulation of GPX4 by ginsenoside Re was key to attenuation of oxidative stress triggered by the hydroxylated dopamine analog 6-OHDA. The antioxidant GPX4 has beneficial effects in a range of pathological conditions, such as neurodegeneration and Alzheimer’s disease [11,31,32,33]. The CCAAT/enhancer-binding protein α was recently found to be responsible for the expression of GPX4 as a transcription factor [34]. However, the molecules associated with the regulation of GPX4 expression remain to be elucidated. Here, we showed for the first time that ginsenoside Re upregulates the expression of GPX4 in SH-SY5Y neuronal cells. Ginsenoside Re was originally shown to protect against dopaminergic neurotoxicity by inducing the dynorphin-mediated κ-opioid receptor in methamphetamine-treated mice [35]. Ginsenoside Re also elicits cytoprotective effects against ultraviolet B-induced oxidative stress in HaCaT keratinocytes by upregulating SOD activity [36]. However, ginsenoside Re only upregulated GPX4, not catalase, SOD1, GR, or GPX1, under our experimental conditions. Therefore, ginsenoside Re may elicit an antioxidant effect, at least in part, by upregulating the key antioxidant gene *GPX4*, thereby preventing excessive phospholipid peroxidation in neurons exposed to oxidative stress.

A number of ginsenosides upregulate multiple antioxidant genes including *GPX1*, *GPX3*, *SOD*, and *heme oxygenase (HO)-1* [17,37,38,39]. Ginsenoside Rb1 elicits an antioxidant effect by upregulating *HO-1* and *SOD* [17,37]. A mixture of ginsenosides Rh1 and Rg2 upregulates *GPX1* [38], while ginsenosides Rb1, Rc, and Re induce *GPX3* [39]. However, ginsenosides have not been previously reported to induce the expression of GPX4 as far as we are aware, although the neuroprotective ability of GPX4 via reduction of phospholipid peroxidation is well-reported [31]. The importance of GPX4 in the cellular redox system was clearly demonstrated in the GPX4-deficient mouse model. These mice undergo embryonic death despite expressing other antioxidant enzymes as normal [11,40]. The antioxidant effect of GPX4 was also demonstrated by the investigation of neuronal cell death evoked by amyloid-β, tert-butyl hydroperoxide, and hydrogen peroxide [41]. Consistent with these previous studies, ginsenoside Re increased resistance of SH-SY5Y cells to oxidative damage. This was attributable to enhanced GPX4 expression because a GPX4-specific inhibitor significantly attenuated the cytoprotective effect of ginsenoside Re. Although (1S3R)-RSL3 alone showed a minor toxic effect on SH-SY5Y cells, the ginsenoside Re-mediated cytoprotection was almost completely reversed in the presence of this inhibitor. Thus, the antioxidant activities of this ginseng-derived compound were associated with its capability to upregulate the intracellular antioxidant protein GPX4. Induction of GPX4 by ginsenoside Re might, therefore, lead to sequestration of lipid peroxides generated during 6-OHDA-induced cellular injury. Indeed, the functional significance of GPX4 in protection against oxidative stress has been well-documented in the kidney, liver, and testis [30,42,43].

A critical finding of this study is that the PI3K/Akt and ERK cascades were associated with ginsenoside Re-mediated induction of the GPX4 gene. The survival signaling cascade defined by PI3K/Akt is critical for the biological activities of ginsenosides Rb1, Re, and Rg1 [26,27,28,29]. This study showed that increases in the levels of phosphorylated Akt and ERK mediated the neuroprotective effects of ginsenoside Re. Ginsenoside Re-induced upregulation of GPX4 expression was abrogated by PI3K inhibitor LY294002 and ERK inhibitor PD98059, but not by p38 inhibitor SB203580 or JNK inhibitor SP600125. Consistent with the actions of PI3K/Akt and ERK as survival factors [44], our findings indicate that ginsenoside Re protects SH-SY5Y cells from oxidative stress triggered by 6-OHDA through upregulation of GPX4 in a process mediated by the PI3K/Akt and ERK signaling cascades.

GPX4 is generally accepted to suppress cellular oxidative stress in vivo and in vitro [10,11,33,41]. On the other hand, ginsenoside Re has been reported to elicit beneficial effects on oxidative stress-mediated neurodegeneration, largely based on its diverse biological actions [28,35]. Therefore, the upregulation of GPX4 expression by ginsenoside Re may alleviate cellular damage by attenuating 6-OHDA-triggered oxidative stress. Consequently, ginsenoside Re has a therapeutically relevant potential for neurodegeneration linked with oxidative stress.

## 4. Materials and Methods

### 4.1. Materials

Ginsenoside Re was provided by Chengdu Biopurify Phytochemicals Ltd. (Chengdu, Sichuan, China). 3-(4,5-dimethylthiazol-2-yl)-2,5-diphenyltetrazolium bromide (MTT) and 6-hydroxydopamine hydrobromide were obtained from Sigma-Aldrich Co. (St. Louis, MO, USA). A monoclonal anti-GPX4 antibody was supplied by Abcam (Cambridge, UK). Polyclonal anti-Akt, anti-phospho-Akt, anti-extracellular signal-regulated kinase (ERK), anti-c-Jun N-terminal kinase (JNK), anti-p38, and anti-phospho-p38 antibodies, and a monoclonal anti-phospho-ERK antibody were supplied by Cell Signaling Technology (Danvers, MA, USA). Monoclonal anti-phospho-JNK and anti-alpha-tubulin antibodies were acquired from Santa Cruz Biotechnology (Dallas, TX, USA). (1S,3R)-RSL3 and TOPscript™ RT DryMIX were obtained from Tocris Bioscience (Bristol, UK) and Enzynomics (Daejeon, Korea), respectively. 2′,7′-Dichlorofluorescin diacetate (H2DCF-DA), LY294002, SP600125, SB203580, and PD98059 were purchased from Calbiochem (La Jolla, CA, USA). TRIzolTM and C11-BODIPY (581/591) were obtained from Thermo Fisher Scientific (Boston, MA, USA).

### 4.2. Cell Culture and Treatment

SH-SY5Y cells (derived from human neuroblastoma) obtained from the Korean Cell Line Bank (Seoul, Korea) were routinely cultured in Minimum Essential Medium with Earle’s Balanced Salts (Hyclone, Logan, UT, USA) containing fetal bovine serum (10%, Life Technologies Corporation, Carlsbad, CA, USA), streptomycin (100 μg/mL), and penicillin (100 U/mL) at 37 °C in an atmosphere of 5% CO_2_ and 95% air. Ginsenoside Re and 6-OHDA were dissolved in dimethyl sulfoxide and distilled water, respectively. Vehicle was used for negative controls. In combined treatment, SH-SY5Y cells were pre-treated in the presence of ginsenoside Re for 9 h before exposure to 6-OHDA.

### 4.3. Immunoblot Analysis

SH-SY5Y cells exposed to each reagent for the indicated duration were washed once with ice-cold phosphate-buffered saline and lysed using a protein extraction solution (PRO-PREP™, iNtRON Biotechnology, Seongnam, Korea). Whole-cell lysates were fractionated by sodium dodecyl sulfate-polyacrylamide gel electrophoresis and transferred to immobilon-P polyvinylidene difluoride membranes (Merck, Darmstadt, Germany). Transferred proteins were blocked using 5% skim milk solution containing 0.1% Tween-20 in Tris-buffered saline (TBS-T) and then reacted with the indicated antibodies at 4 °C. Following brief washing with TBS-T, the membranes were incubated with a peroxidase-conjugated antibody at ambient temperature for 2 h. Following thorough washing with TBS-T, reactive proteins were visualized using WesternBright ECL (Advansta Inc., Menlo Park, CA, USA).

### 4.4. Cytotoxicity Assay

The cytotoxic effect of each reagent was assessed by both the MTT assay and LDH release. For the MTT assay, SH-SY5Y cells were seeded in 24-well plates (4 × 10^4^ cells/well) and then treated with various reagents alone or in combination for the indicated durations. The cells were then incubated for an additional 2 h in culture medium containing MTT solution (final 0.1 mg/mL concentration). Following removal of the medium, the formazan crystals which formed in living cells were dissolved using acidified isopropanol solution. The absorbance of the solution was then measured at 570 nm using a Multiskan™ GO Microplate Spectrophotometer (Thermo Scientific, Waltham, MA, USA). For the LDH release assay, the cells were treated with indicated reagents as described for the MTT assay, and the amount of LDH secreted into the culture medium was determined by spectrometrically measuring the absorbance of the colored product at 490 nm using a kit (CytoTox 96 Non-radioactive Cytotoxicity Assay, Promega, Madison, WI, USA).

### 4.5. Measurement of Intracellular ROS

SH-SY5Y cells were plated in 6-well plates (1.6 × 10^5^ cells/well) and then exposed to the indicated reagents. Following exposure to 50 μM H2DCF-DA for a final 30 min, green fluorescence, which was proportionate to the level of intracellular ROS, was measured using an Eclipse Ti2 fluorescence microscope (Nikon, Tokyo, Japan).

### 4.6. Real-Time PCR Analysis

Total RNA was extracted using TRIzol (Thermo Fisher Scientific) from SH-SY5Y cells treated with the indicated reagent for the indicated duration, and then reverse-transcribed into cDNA by a kit (TOPscript™ RT DryMIX, Enzynomics). Real-time PCR was performed with equal amounts of cDNA using a PCR Master Mix (Mix (Solgent, Daejeon, Korea) and primers. The PCR conditions are as follows: initial denaturation for 15 min at 95 °C, followed by 40 cycles of 15 s at 95 °C, 15 s at 56 °C, and 30 s at 72 °C using a Roche Diagnostics LightCycler^®^ 96 (Basel, Switzerland). The primers used: *SOD1*, 5′-GAGACTTGGGCAATGTGAC-3′ and 5′-ACCTTTGCCCAAGTCATCTG-3′; *SOD2*, 5′-ACAGGCCTTATTCCACTGCT-3′ and 5′-CAGCATAACGATCGTGGTTT-3′; *GR*, 5′-CTGAAGTTCTCCCAGGTCAAG-3′ and 5′-AGGCAGTCAACATCTGGAATC-3′; *CAT*, 5′-TAAGACTGACCAGGGCATC-3′ and 5′-CAAACCTTGGTGAGATCGAA-3′; *GPX1*, 5′-CGGGACTACACCCAGATGA-3′ and 5′-TCTTGGCGTTCTCCTGATG-3′; *GPX4*, 5′-GGGCTACAACGTCAAATTCG-3′ and 5′-TCCACTTGATGGCATTTCCC-3′; and *RPS18*, 5′-TGCGAGTACTCAACACCAAC-3′ and 5′-GTCTGCTTTCCTCAACACCA-3′.

### 4.7. Determination of Lipid Peroxidation

SH-SY5Y cells plated in 6-well plates (1.6 × 105 cells/well) were exposed to reagent for the indicated duration. The cells were then incubated with 1 μM C11-BODIPY (581/591) (Thermo Fisher Scientific) for a final 30 min. Fluorescence of C11-BODIPY, which corresponded to the level of lipid peroxidation, was measured using an Eclipse Ti2 fluorescence microscope (Nikon).

### 4.8. Statistical Analysis

Data were analyzed using SigmaPlot 10 software (Systat, Chicago, IL, USA) and are expressed as mean ± standard error (SE). The statistical significance was determined by an unpaired t-test with Welch’s correction. *p* < 0.05 was considered statistically significant.

## Figures and Tables

**Figure 1 molecules-25-00188-f001:**
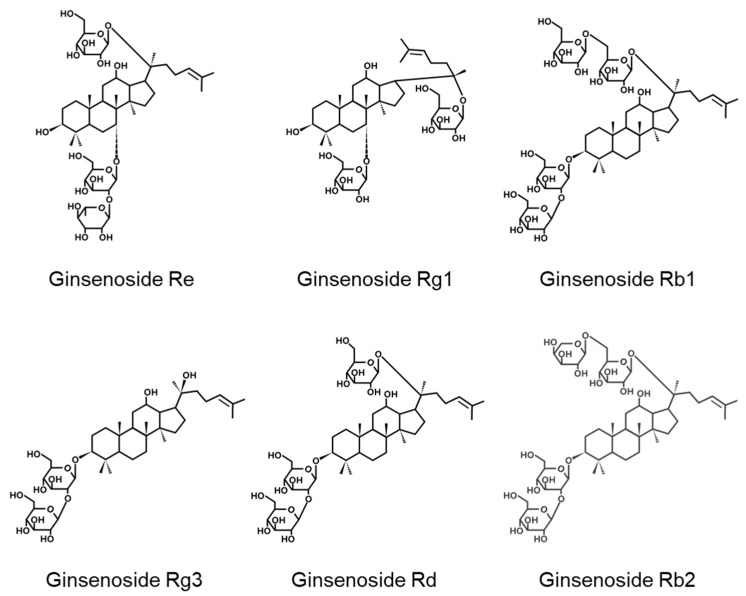
Chemical structures of ginsenosides.

**Figure 2 molecules-25-00188-f002:**
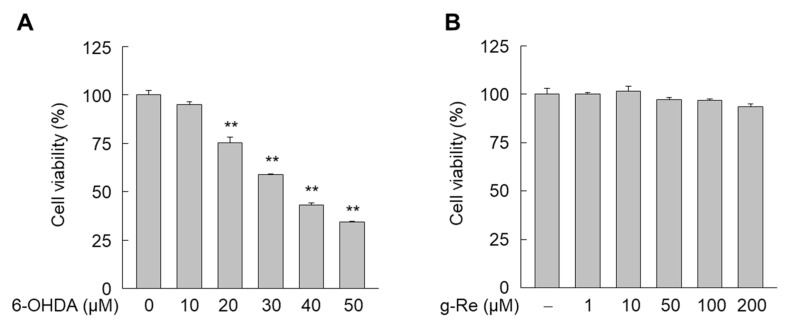
Effect of 6-Hydroxydopamine (6-OHDA) and ginsenoside Re (g-Re) on the viability of SH-SY5Y cells. Cells were treated with various concentrations of 6-OHDA or g-Re for 24 h, and cell viability was determined by the 3-(4,5-dimethylthiazol-2-yl)-2,5-diphenyltetrazolium bromide (MTT) assay. ** *p* < 0.01 compared to untreated group.

**Figure 3 molecules-25-00188-f003:**
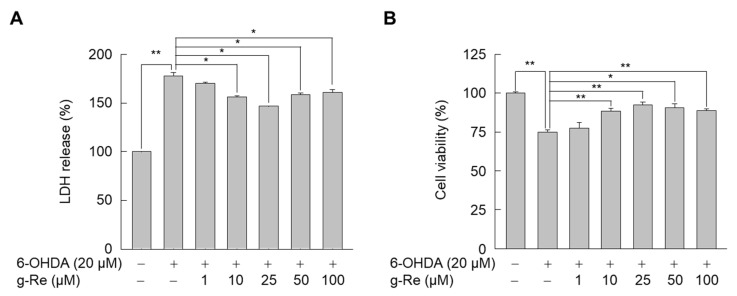
Effect of ginsenoside Re (g-Re) on 6-OHDA-triggered cellular damage in SH-SY5Y cells. Cells were pretreated with the indicated concentrations of g-Re for 9 h and then exposed to 6-OHDA for 24 h. The (**A**) lactate dehydrogenase (LDH) release assay and (**B**) MTT assay were performed to investigate cellular injury. Data are presented as mean ± SE (n = 3). * *p* < 0.05, ** *p* < 0.01.

**Figure 4 molecules-25-00188-f004:**
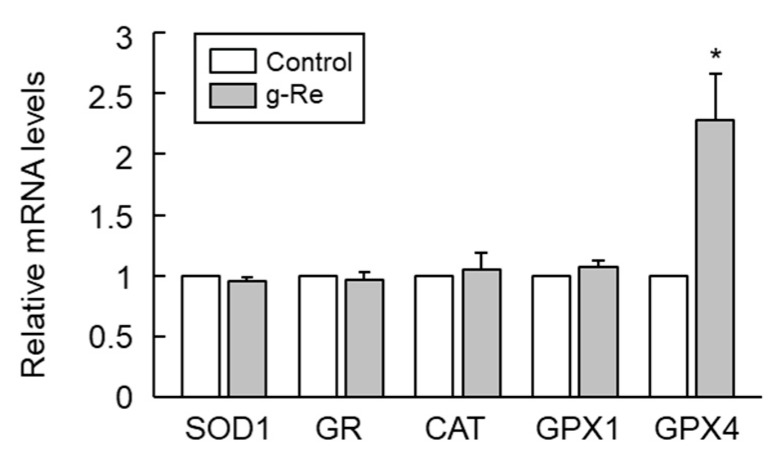
Effect of ginsenoside Re (g-Re) on the expression of antioxidant genes in SH-SY5Y cells. Cells were treated with or without 25 μM g-Re for 9 h. Total RNA was extracted, and mRNA levels of the indicated genes were analyzed by real-time PCR. Results are expressed as mean ± SE (n = 3). * *p* < 0.05.

**Figure 5 molecules-25-00188-f005:**
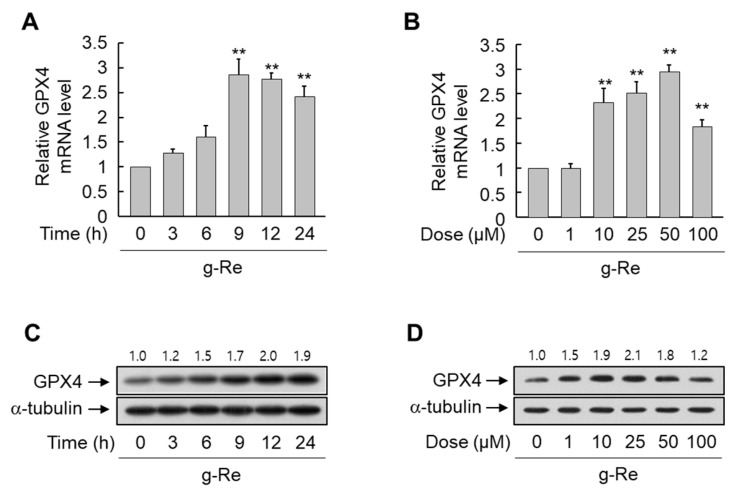
Effects of ginsenoside Re (g-Re) on the expression of glutathione peroxidase 4 (GPX4) in SH-SY5Y cells. (**A** and **C**) Cells were exposed to 25 μM g-Re for the indicated durations. (**B**,**D**) Cells were treated with the indicated concentrations of g-Re for 9 h (**B**) and 24 h (**D**). The mRNA and protein levels of GPX4 were analyzed by real-time PCR (**A**,**B**) and immunoblotting (**C**,**D**), respectively. Results are expressed as mean ± SE (n = 3). RPS18 and α-tubulin were used as internal controls for real-time PCR and immunoblotting, respectively. ** *p* < 0.01 compared with the untreated group.

**Figure 6 molecules-25-00188-f006:**
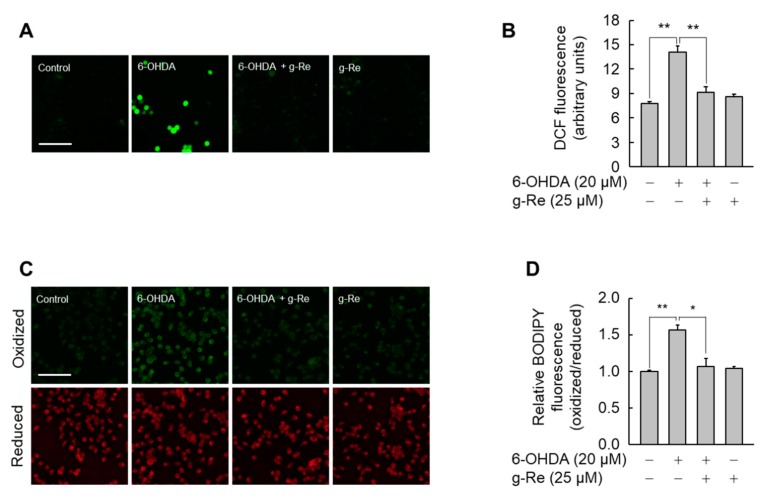
Effect of ginsenoside Re (g-Re) on 6-OHDA-induced reactive oxygen species (ROS) production and lipid peroxidation. SH-SY5Y cells pretreated with g-Re for 9 h were incubated with or without 6-OHDA. (**A**,**B**) Following incubation for 24 h, cells were further incubated in medium containing 50 μM 2′,7′-Dichlorofluorescin diacetate (H2DCF-DA) for 30 min. The intracellular ROS level was determined by fluorescence microscopy (**A**), and the fluorescence intensities were quantified (**B**). (**C**,**D**) Lipid peroxidation was investigated in cells incubated with 1 μM C11-BODIPY for the final 30 min. Fluorescence of C11-BODIPY, which corresponded to the level of lipid peroxidation, was measured by fluorescence microscopy (**C**). The intensities of fluorescence (**B**) and oxidized/reduced C11-BODIPY (**D**) are presented as mean ± SE (n = 3). Scale bars = 100 μm. * *p* < 0.05, ** *p* < 0.01.

**Figure 7 molecules-25-00188-f007:**
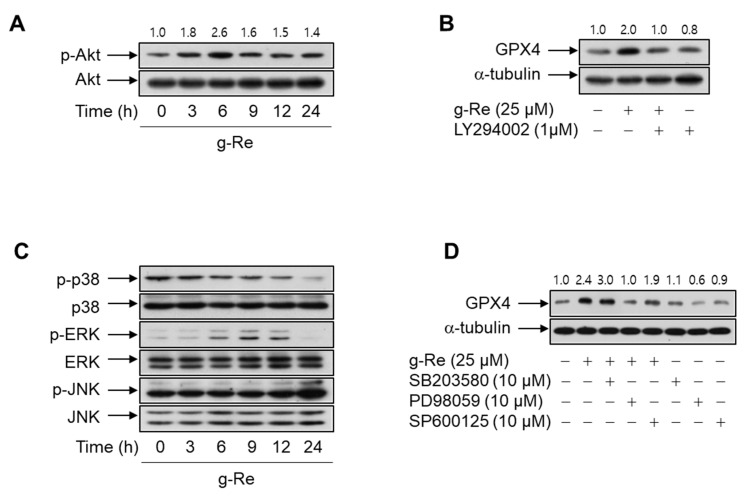
Roles of the phosphoinositide 3-kinase (PI3K)/Akt pathway and mitogen-activated protein kinase cascades in ginsenoside Re (g-Re)-induced upregulation of GPX4. (**A**,**C**) SH-SY5Y cells were treated with 25 μM of g-Re for the indicated durations. An aliquot of the whole-cell lysate was immunoblotted with activation-specific antibodies, while total kinase levels were analyzed in parallel immunoblots. (**B**,**D**) SH-SY5Y cells pretreated with vehicle (Dimethyl sulfoxide), LY294002, SB203580, PD98059, or SP600125 for 1 h were incubated with or without g-Re for 24 h. The level of GPX4 protein was assessed by Western blot analysis. α-Tubulin was used as an internal control for immunoblotting.

**Figure 8 molecules-25-00188-f008:**
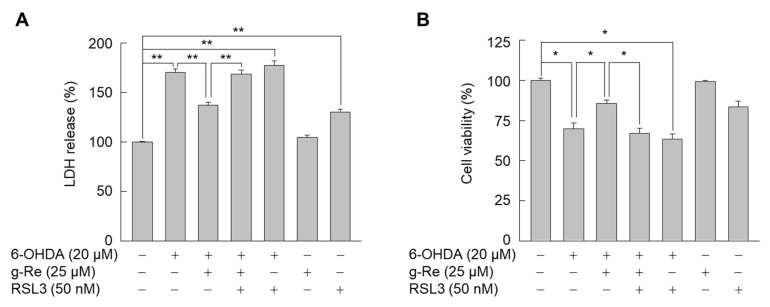
Effects of GPX4 inhibition on ginsenoside Re (g-Re)-mediated suppression of 6-OHDA-induced cellular damage. (**A**,**B**) SH-SY5Y cells pretreated with or without (1S,3R)-RSL3 (RSL3) were incubated in the presence or absence of g-Re for 9 h and then treated with or without 6-OHDA for 24 h. Cells were subjected to the LDH release assay (**A**) and MTT assay (**B**). Data are presented as mean ± SE (n = 3). * *p* < 0.05, ** *p* < 0.01.

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
