# Peer review of "Ginsenoside Re Mitigates 6-Hydroxydopamine-Induced Oxidative Stress through Upregulation of GPX4"

_molecules, 2020, doi:10.3390/molecules25010188_

Round 1

Reviewer 1 Report

The present paper reports data on the protective efficacy of ginsenoside Re on SH_SY5Y cells stressed by 6-OHDA. Results are potentially interesting but there are some questions deserving an answer. What is the main difference among ginsenosides? Is it easy to isolate them from the plant or they are synthetized? Are separate ginsenosides more active than a mixture? Have they been tested in the same model (separate and mixture)?

2.2 it is correct to assume that the expression of antioxidant proteins regulated by ginsenoside Re was tested only in un-stressed cell culture? Is this regulation independent from stress? What happens under stress? Are the same proteins regulated? It might be that under stress ginsenosides are able to regulate more or different genes. The protective effect might be due to a complex mechanism not limited to up-regulation of GPX4. Although results reported in section 2.5 support a relevant role of GPX4 do not exclude additional effect. Looking at fig.7 GPX4 appears relevant but not the only one.

a list of abbreviation might be added

Reviewer 2 Report

This article details the biological studies of Ginsenoside Re in a 6-OHDA neuroprotection model on cell for Parkinson disease.

The authors are introducing the article by reviewing the different causes that can induce Parkinson disease and followed by introducing biological activities of Ginsenosides, in particular anti-oxidant activities that might suppress anti-oxidant stress cause for Parkinson disease.

  Some Ginsenoside were found to have this activity profile in the literature, and then the authors focus their attention on Ginsenosides Re. At this stage I was surprised to see no real scientific base leaded to the study of ginsenoside Re. The only explanation was “the antioxidant effect of ginsenoside Re in the neuronal system has not been fully explored”.

I am not a specialist of ginsenoside, but as a chemist, I was surprised no figure showing the structures of Ginsenosides (particularly for Re) was furnished. I strongly suggest to show a figure with all ginsenoside reported by the author in the texte (mainly Re, Rg1, Rb1, Rb2, Rd, Rg3). This is of primary importance since “Molecules” is read mainly by chemists I believe, and also, a general overview would be needed to understand maybe why ginsenoside Re was specifically studied.

To this last comment I would also add that all ginsenosides have a similar structure, the only difference being in the nature of the saccharides. I do not believe the antioxidant activity comes from sugar moiety, therefore studying a more specific ginsenoside among all other is not very pertinent excepted if a scientific study could prove the contrary.

Moreover, antioxidant activity is a very common activity for many compounds and natural products (if I search resveratrol and 6-OHDA I rapidly see some article about a similar activity: Brain Res. 2010 Apr 30;1328:139-51.), and by consequence having an activity on oxidative stress in Parkinson disease model is not unexpected. The author did not show Ginsenoside Re is more specific to GPX4.

The study would have been more interesting if it have been shown that ginsenosides had a in vivo activity for neuroprotection, since the plant from where they are isolated is largely consumed.

Reviewer 3 Report

The manuscript “Ginsenoside Re Mitigates 6-hydroxydopamine-Induced Oxidative Stress through Upregulation of GPX4” describes the neuroprotective effects of the ginsenoside Re against neurotoxicity induced by 6OHDA in SH-SY5Y cells. The main result of the work is that the neuroprotection is elicited by the upregulation of the antioxidant enzyme GPX4.

The results obtained regarding the role of GPX4 are remarkable considering the interest in the neuroprotective properties of ginsenoside. However, I have some recommendation and comments; some technical details should be expanded and clarified to ensure that readers understand exactly what the authors studied

The purpose of the work is not clear: it seems a test to evaluate the protection of Re against neurotoxicity of 6-OHDA. 6-OHDA is a well-known toxin able to induce Parkinson's Disease-like phenotype. It is also found naturally in the human urine and is derived from dopamine, in fact, the 6OHDA level increase in patients treated with dopamine. The introduction section should be modified to better explain the purpose of the work to improve the meaning of the cellular model used, that is simple. A model should be able to mimic disease or a pathological process related to the human condition. I suppose the aims should be to test the neuroprotective effects of Re in an in vitro model of PD induced by 6-OHDA. A suggestion for future studies: SH-SY5Y cells can be easily differentiated in dopaminergic-like neurons using retinoic acid for a few days. The usage of this model can improve the quality of the message the authors want to deliver. Which are the solvents of Re and 6-OHDA? They should be cited in the Material and Methods, in consideration of their final concentration in the negative control (especially for a solvent that can cause slight toxicity). How was chosen the 6-OHDA concentration for the treatment? Could the authors explain the reasons for the measurement of mRNA expression at 9 hours? Considering the toxic treatment of 24 hours. The expression of different genes could vary of hours and days after toxic stimuli [https://doi.org/10.3390/ijms20092224]. The same 0-3-6-9-12-24 hours should be done for all the genes, for a complete analysis. The introduction clearly stated the interest of the author in GPX4, but the experimental protocol used should demonstrate the reasons for this choice. Immunoblotting should be quantified for a better evaluation of the results, as expression level vs. negative control / 0 hours = 1. Lines 119-120: could the authors specify if Re was used in a single treatment or to counteract the toxic effects of 6OHDA? It should be more interesting to understand the role of Re also during the toxic stimuli. Paragraph 2.4 should be moved after paragraph 2.2, to follow the reasoning of the antioxidant effect of Re. The detailed study of the molecular mechanisms involved can follow. Line 153-15: correct 25 M in 25 uM. The work needs to better clarify or at least a paragraph regarding experimental design should be added. Before the authors describe co-treatment Re-6OHDA, then only Re, then pre-treatment Re followed by 6OHDA, without any clear description of the protocol in the Materials and Methods or a brief introduction of each paragraph of the Results section. Figure 6: usually a positive control (H2O2) is suggested in the case of measurement of ROS, to set up the maximum level of oxidative stress. Figure 6D: correct BODIFY in BODIPY in the name of y-axes of the graph. Figure 7A-B: the results are interesting and show the impact of GPX4 on neuroprotection properties of Re. The limitation of the description of the results is that RSL3 is toxic for the cells and the inhibition of basal level of GPX4 is deleterious for the cells. It could be considered an important point to discuss, as it seems to represent an indispensable component of basal cell protection. Are the authors sure that in Figure 7B the damage induced by RSL3 isn’t significant? Line 275-285-286-304: correct the superscript and subscript.

Reviewer 4 Report

Lee et al the protective effect of Ginsenoside Re in 6- OHDA-induced oxidative stress SH-SY5Y cells. They demonstrated that cell exposure to Ginsenoside Re treatment led to the increase of GPX4 mRNA and protein level and ix concomitant to the induction of phosphoinositide 3-kinase and extracellular signal-regulated kinase.The resultats are of interest, because of the possible impact of the molecule in PD.

However , in the present format, the manuscript needs revision

Major points

Introduction:

Lines 36-48: The authors are encouraged to state directly that 6-OHDA is a  PD reference molecule and give the appropriate references.

Lines 44-53: As stated, oxidative stress defence seems to be limited to antioxidant enzyme activity, whereas non enzymatic molecules such as glutathione is also involved.This point is also discussed as a major point for the results section

Results:

The choice of the cell line used must by explained The authors did not statedthe choice of the concentration of 6-OHDA or Ginsenoside Re used (depending on results obtained by MTT assay or maximum of GPX4 mRNA level increase?) The authors are encouraged to present LDH and MTT experiment performed with 6- OHDA, or to give the IC50 obtained after cell compound treatment (notably for Fig 7 analysis) Since the 6-OHDA treatment increased ROS production, ginsenoside Re increased GPX4 level (both at the mRNA and protein level), do the authors quantify the level of GSH  in their assays? In addition, the authors are encouraged to present kinetic of ROS generation, during cell 6-OHDA treatment and during 6-OHDA combined to Ginsenoside Re ,since they tested ROS generation only after cell exposure for 24h-treatment delay. It is generally admitted that ROS generation is a rapid process starting with cell exposition to drug or molecule. -The link between PI3K pathway and compound treatment might be stated before the results obtained. There is no clear explanation why the authors studied this pathway in this section, even though this point is discussed thereafter Quantification of the bands obtained by western blotting must be given The analysis of figure 7 is confusing: What do the authors want to demonstrated : a protective effect id est antioxidant effect of ginsenoside or that GPX4 inhibitor and Ginsenoside RE behave a different protein/patway target?

Minor points

Line 54 : modify or prevent?

Line 76:” remarquably” is excessive

Round 2

Reviewer 1 Report

The authors responded to all the questions opened but it would be nice to see some of them in the new version specifically the response to the point 2. 2.2

Author Response

Thanks to your constructive comments. Please see the attached file.

Reviewer 3 Report

All the concerns from the previous version of the submitted manuscripts were addressed.

Author Response

Thanks to your constructive comments.

Reviewer 4 Report

Dear,

The manuscript could be improved by the following points.

Line 105, suggested change: "Treatment applied induced only changes at the mRNA level of GPX4 (figure 4)".

Lines 81-84: "Next, we assessed the effects of 81 ginsenoside Re on 6-OHDA-triggered cellular damage by determining the level of lactate 82 dehydrogenase (LDH) released into the medium and MTT-based cell viability in SH-SY5Y cells. 83 6-OHDA significantly increased LDH release into the medium as reported previously [3]. However…."

The manuscript will be improved by adding the figure (in author’s comments) which summarises cell viability after cell exposure to 6-OHDA. Figure should be presented as figure 2A. Moreover, it is suggested to add the following sentences:

“In previous experiments, we measured the effect of 6-OHDA on SH-SY5Y cell viability (Figure 2A). Cell viability decreased when concentration of 6-OHDA increased. Since our purpose consists to demonstrate the protective effect of ginsenoside Re on 6-OHDA cell survival, SH-SY5Y cells were exposed thereafter to 20 µM 6-OHDA, which treatment decreases cell viability up to 30%”.

Line 105: suggested sentence ‘Treatment applied induced only changes at the mRNA level of GPX4 (figure 4)”.

Line 154: suggested sentence: "Because diverse  ginsenoside derivatives treatment [40–43] activate PI3K/Akt signaling pathway, as a cell survival signalling pathway [26], we assessed whether it is involved in cell survival of 6-OHDA treated cells.

Line 231: Suggested sentence: Although RSL3 alone showed a minor toxic effect on SH-SY5Y cells, the ginsenoside Re-mediated cytoprotection was almost completely reversed in the presence of this inhibitor.

Line 275: suggested changes:”Ginsenoside Re and 6-OHDA were dissolved in DMSO and distilled water, respectively. Vehicle was used for negative controls. In combined treatment, SH-SY5Y cells were pre-treated in the presence of ginsenoside Re for 9 h, before exposure to 6-OHDA.”

Best Regards
